# Does the Heart Fall Asleep?—Diurnal Variations in Heart Rate Variability in Patients with Disorders of Consciousness

**DOI:** 10.3390/brainsci12030375

**Published:** 2022-03-11

**Authors:** Monika Angerer, Frank H. Wilhelm, Michael Liedlgruber, Gerald Pichler, Birgit Angerer, Monika Scarpatetti, Christine Blume, Manuel Schabus

**Affiliations:** 1Laboratory for Sleep, Cognition and Consciousness Research, Department of Psychology, University of Salzburg, 5020 Salzburg, Austria; monika.angerer@plus.ac.at; 2Centre for Cognitive Neuroscience Salzburg (CCNS), University of Salzburg, 5020 Salzburg, Austria; 3Division of Clinical Psychology and Psychopathology, Department of Psychology, University of Salzburg, 5020 Salzburg, Austria; frank.wilhelm@plus.ac.at (F.H.W.); michael.liedlgruber@plus.ac.at (M.L.); 4Apallic Care Unit, Albert Schweitzer Hospital, Geriatric Health Care Centres of the City of Graz, 8020 Graz, Austria; gerald.pichler@stadt.graz.at (G.P.); monika.scarpatetti@stadt.graz.at (M.S.); 5Private Practice for General Medicine and Neurology, 8430 Leibnitz, Austria; leibnitz@braindoc.at; 6Centre for Chronobiology, Psychiatric Hospital of the University of Basel, 4002 Basel, Switzerland; christine.blume@unibas.ch; 7Transfaculty Research Platform Molecular and Cognitive Neurosciences, University of Basel, 4055 Basel, Switzerland

**Keywords:** disorders of consciousness, brain injury, ECG, heart rate, heart rate variability, diurnal variation

## Abstract

The current study investigated heart rate (HR) and heart rate variability (HRV) across day and night in patients with disorders of consciousness (DOC). We recorded 24-h electrocardiography in 26 patients with DOC (i.e., unresponsive wakefulness syndrome (UWS; *n* = 16) and (exit) minimally conscious state ((E)MCS; *n* = 10)). To examine diurnal variations, HR and HRV indices in the time, frequency, and entropy domains were computed for periods of clear day- (forenoon: 8 a.m.–2 p.m.; afternoon: 2 p.m.–8 p.m.) and nighttime (11 p.m.–5 a.m.). The results indicate that patients’ interbeat intervals (IBIs) were larger during the night than during the day, indicating HR slowing. The patients in UWS showed larger IBIs compared to the patients in (E)MCS, and the patients with non-traumatic brain injury showed lower HRV entropy than the patients with traumatic brain injury. Additionally, higher HRV entropy was associated with higher EEG entropy during the night. Thus, cardiac activity varies with a diurnal pattern in patients with DOC and can differentiate between patients’ diagnoses and etiologies. Moreover, the interaction of heart and brain appears to follow a diurnal rhythm. Thus, HR and HRV seem to mirror the integrity of brain functioning and, consequently, might serve as supplementary measures for improving the validity of assessments in patients with DOC.

## 1. Introduction

Severe brain injury can cause coma, and, upon recovery, changes in consciousness often persist. These states are categorized under the term ‘disorders of consciousness’ (DOC). In a simplified approach, two major components are thought to be necessary for consciousness: wakefulness (i.e., the level of arousal) and awareness of the environment and the self (i.e., contents of consciousness) [1]. In patients living with DOC, wakefulness is preserved but awareness is only intermittently present or completely absent. More specifically, while patients with an unresponsive wakefulness syndrome (UWS) show some return of arousal (i.e., phases of sleep (closed eyes) and wakefulness (open eyes)) without signs of awareness during behavioral assessment, patients in a minimally conscious state (MCS) show inconsistent but reproducible signs of awareness that can be differentiated from reflexive behavior (e.g., response to commands, visual pursuit, intentional communication) [2,3]. If patients can communicate functionally and use objects adequately, their state is denoted as emergence from minimally conscious state (EMCS) [4]. Thus, while patients with UWS are assumed to be unconscious, patients in MCS and EMCS are assumed to be (at least minimally) conscious.

Clinical diagnoses are usually based on observations of the patients’ behavior using, for instance, the Glasgow Coma Scale [5] for acute situations or the Coma Recovery Scale-Revised (CRS-R) [6] for tracing their development during recovery. Unfortunately, behavioral assessments involve the risk of underestimating the level of consciousness [7]. This is because patients may be unable to respond behaviorally (and thus give evidence of their consciousness), for example, due to sensory or motor impairments. Finally, the fluctuating levels of consciousness carry the risk of examinations taking place during windows of unconsciousness [8]. Consequently, the absence of evidence for consciousness must not be mistaken for evidence of its absence. Hence, distinguishing between UWS and (E)MCS continues to be a challenge in clinical practice, and the rate of misdiagnoses is high (i.e., ~40%) [7] when comparing the medical consensus to the results obtained through clinical scales [9]. Approaches based on neuroimaging methods, such as functional magnetic resonance imaging (fMRI) or electroencephalography (EEG), have been used as additional tools to improve the validity of DOC diagnoses [10,11]. However, they require special expertise, come with high time and financial requirements, and often rely on tasks that are still challenging for patients [12]. Therefore, researchers and clinicians are looking for alternative or adjunct measures that (i) do not rely on patients’ behavioral responses, (ii) are time- and cost-efficient, (iii) and can be easily applied at beside. Lastly (iv), they should be useable during longer-term recordings, thereby taking the problem of fluctuating consciousness levels over the circadian day into account.

Among these measures, heart rate (HR) and heart rate variability (HRV) have been suggested to fulfill these criteria [13]. Specifically, HR indicates the average time interval between adjacent heartbeats (i.e., interbeat interval; IBI), while HRV quantifies the variability in these time intervals. These variations occur at different frequencies and reflect dynamics of the autonomic nervous system (ANS) regulation, being related to sympathetic and parasympathetic activity, breathing, thermo-, and blood pressure regulation, as well as changes in the vasomotor and renin-angiotensin system [14] (cf. Box 1). Furthermore, HR/HRV may mirror the interaction between the (injured) brain and the heart and thus represent a ‘peripheral’ window to ‘central’ functioning. The neural structure that enables the bidirectional communication of the heart and central nervous system has been described as the central autonomic network, which has been suggested to be involved in cognitive, emotional, and autonomic regulation, and linked to HR/HRV and cognitive performance [15].

In patients with severe brain injury, a decrease in HRV parameters in the time and frequency domain has been associated with a worse clinical outcome [16,17]. When looking at HRV entropy, it has been shown that patients with UWS show lower approximate entropy (ApEn) values than healthy controls [18]. Furthermore, higher complexity values, which were associated with higher CRS-R scores, were observed in patients in MCS as compared to patients with UWS [13]. Interestingly, no differences between patients with UWS and healthy controls have been found in any of the linear parameters (i.e., root mean square of successive differences between adjacent heartbeats (RMSSD), standard deviation of IBIs (SDRR), ratio between low and high frequencies (LF/HF ratio)) [18]. Furthermore, cognitive processes associated with ANS responses elicited by nociceptive stimuli have been identified in patients with MCS but not in patients with UWS. Specifically, following a repetitive nociceptive laser stimulation, patients in MCS showed a higher HR and SDRR as compared to before the stimulation, whereas patients with UWS did not show a significant change [19].

Importantly, like other physiological signals (such as body temperature and hormone secretion), HR/HRV in healthy populations shows strong alterations between sleep and wakefulness [20,21,22], so-called circadian variations (i.e., rhythms with a period length of approximately 24 h). In patients with severe brain injury, it has been shown that circadian temperature, melatonin, or motoric activity rhythms often deviate from the healthy norm and that a better integrity of the patients’ circadian rhythm is associated with a better clinical state [8,23,24]. Unfortunately, there is only limited research on how day vs. night variation affects the HR/HRV of patients with DOC. In patients with UWS, it has been shown that 6/11 patients had a higher HR during the night (i.e., 9 p.m.–5 a.m.) as compared to the day (5 a.m.–9 p.m.) [25]. Until now, it remains unclear to what degree the existence of diurnal variation in HR/HRV is linked to the patients’ clinical state.

Thus, our aim was to explore HR/HRV in patients with DOC in the time, frequency, and entropy domains. Specifically, as light is the most important zeitgeber for the internal biological clock [26], we also took the lighting conditions in the patients’ room into account and analyzed data from periods of definite day- and nighttime. Further, we investigated whether it is possible to differentiate between DOC diagnoses based on patients’ cardiac characteristics, and whether there is an association between the patients’ heart and brain activity.

Box 1HRV parameters in the time, frequency, and entropy domains.Methodologically, HRV can be evaluated in the frequency, time, and entropy domains. In the frequency domain, researchers usually compute a fast Fourier transform for three specific frequency bands that can be related to functions of the autonomic nervous system: high (HF; 0.15–0.4 Hz), low (LF; 0.04–0.15 Hz), and very low frequency (VLF; 0.003–0.04 Hz). While the HF band represents parasympathetic activity, the LF band reflects
both parasympathetic and sympathetic activity and is particularly related to blood pressure regulation. The VLF band is mainly related to the thermoregulation, vasomotor, and renin–angiotensin system. In the time domain, variation in IBIs can be quantified by different parameters, such as the root mean square of successive differences between adjacent heartbeats (RMSSD), which is used to estimate variation mainly related to the HF band (pointing to parasympathetic activity; for other time-domain parameters, see Ref. [14]).
By using non-linear methods (i.e., entropy domain), the complexity and irregularity of the IBI signal can be investigated. Approximate entropy (ApEn), detrended fluctuation analysis scaling exponent (DfaAlpha), Hurst exponent (Hurst), and sample entropy (SampEn) are amongst the most commonly used non-linear methods for HRV analysis [14,27,28]. Besides HRV, heart rate (HR) also provides information
about autonomic nervous system regulation. Specifically, HR is a basic measure of overall cardiovascular arousal, which is influenced by both sympathetic activation and parasympathetic withdrawal.

## 2. Materials and Methods

### 2.1. Patients

From a total of 38 non-sedated and spontaneously breathing patients with DOC, nine patients had to be excluded from the analyses due to severe cardiac arrhythmias. Three further patients were excluded because they were exposed to constant 24-h light. Thus, 26 patients (8 women) between 16 and 80 years old (*mean* = 50.29, *sd* = 19.96) with different etiologies (traumatic brain injury (TBI): *n* = 11, non-traumatic brain injury (NTBI): *n* = 15) from clinics in Austria (*n* = 12) and Belgium (*n* = 14) were included in the analyses. The patients’ behavioral state was assessed with the CRS-R [6]. While 16 patients were diagnosed with UWS, nine patients were in an MCS, and one was in an EMCS. Patients with MCS and EMCS were combined for statistical analyses, which allows us to analyze differences between unconscious patients with UWS and (at least minimally) conscious patients in (E)MCS. Note that the data of this study were collected in a broader research context with the aim to quantify sleep and circadian parameters. Specifically, data of both patient samples (i.e., Austria and Belgium) have been used in two previous publications, where we studied diurnal variations in EEG parameters [29], as well as circadian variations in skin temperature [30] of patients with DOC. However, these studies did not focus on diurnal variations in electrocardiogram (ECG) data. The studies have been approved by the local ethics committees and informed consent was obtained from the patients’ legal representatives. For details on the study sample, see Table 1.

### 2.2. Study Protocol

#### 2.2.1. Austria

In Austria, the study protocol comprised two within-subject conditions, namely the (i) habitual light (HL) and (ii) dynamic daylight (DDL) condition. Each condition lasted one week, and orders were randomized. While patients were in a room with standard clinic room lighting in the HL condition, patients were in a room with a ‘biologically effective’ room light (i.e., light intensity and spectrum that mimics natural daylight) in the DDL condition. During both weeks, skin temperature and actimetry were assessed continuously. ECG data were acquired at the beginning of each study condition (i.e., with a minimum recording duration of 1.5 days and a maximum recording duration of 4.9 days). The patients’ behavioral repertoire was assessed twice with the CRS-R at the end of each week, once in the morning and once in the afternoon, in all patients (*n* = 12). The data from the DDL condition are beyond the scope of this study. For the current analyses, the CRS-R assessments from the HL condition as well as 24-h continuous ECG data recorded at the beginning of the HL condition (i.e., starting on the first or second day) were analyzed (for more details on the study protocol, see Ref. [30]).

#### 2.2.2. Belgium

In Belgium, a 24-h polysomnography (PSG; including EEG, ECG, electromyogram (EMG), electrooculogram (EOG), and respiration) was performed in the patients’ usual clinical environment with standard clinic room lighting (i.e., HL). The CRS-R was performed twice (i.e., before and after the recording) in nine and once in five of the 14 patients.

For the current analyses, the data of the Belgian patient sample were combined with the HL data of the Austrian patient sample as lighting conditions were comparable in these two datasets. Thus, we are looking at HR/HRV under standard clinic room lighting conditions.

### 2.3. Behavioral Assessment and Data Analysis

#### 2.3.1. Coma Recovery Scale-Revised

The behavioral state of the patients was assessed with the CRS-R [6], which is composed of six subscales reflecting auditory, visual, motor, oromotor, communication and arousal functions with a total of 23 items. While the lowest threshold item on each subscale represents reflexive behavior, the highest threshold item indicates cognitively mediated behavior. The CRS-R is performed in a hierarchical manner, which means that the examiner starts with the highest item of each subscale and moves down the scale until the patient’s response meets the criteria for one item. When two CRS-R assessments were available, we used the CRS-R assessment where the patients showed the highest behavioral reactivity (i.e., the best diagnosis or highest sum score) as this is thought to best represent the ‘true’ state of the patient. The highest CRS-R score and diagnosis of each patient is shown in Table 1 and the subscale scores in Appendix A.

#### 2.3.2. Electrocardiography

For recording ECG in the patient sample from Austria, an ambulatory three-channel ECG device (eMotion Faros 180°, Mega Electronics Ltd., Kuopio, Finland) with self-adhesive Ambu^®^ BlueSensor SP electrodes (AMBU A/S, Ballerup, Denmark) was used. Two electrodes were placed in an infraclavicular position on the right and left body side, and one on a rib on the lower left thoracic wall. The sampling rate was 1000 Hz.

In the patient sample from Belgium, the ECG was recorded as part of the PSG. For PSG recordings, BrainProducts amplifiers (BrainProducts, Gilching, Germany) were used. All PSG signals were recorded with a sampling rate of 500 Hz. For ECG recordings, two goldcup electrodes were used. One electrode was affixed in an infraclavicular position on the right body side and one on a rib on the lower left thoracic wall using EC2 electrode gel (Astro-Med^®^ Inc., West Warwick, RI, USA).

#### 2.3.3. Heart Rate and Heart Rate Variability

To explore diurnal variations in HR/HRV, we used continuous 24-h ECG data from each patient and divided the ECG recording into periods of clear daytime (i.e., forenoon: 8 a.m.–2 p.m., afternoon: 2 p.m.–8 p.m.) and nighttime (i.e., 11 p.m.–5 a.m.) using BrainVision Analyzer 2.0 Software (Brain Products GmbH, Gilching, Germany). Importantly, all three periods were of equal length (i.e., 6 h). Periods with conditions of twilight (i.e., dawn: 5–8 a.m., dusk: 8–11 p.m.) were excluded from the analyses.

HRV analyses were conducted in ANSLAB 2.6 [31]. Accurate automatic R peak detections (obtained with the software’s algorithms) were carefully visually checked for the entire data of each patient and corrected whenever necessary. In a second and important preprocessing step, the IBI data were visually inspected for arrhythmias (e.g., ectopic beats) as single spikes in the IBI signal can seriously distort the spectral estimates of HRV [32]. While single spikes were interpolated with the software’s algorithm (i.e., mean of the adjacent IBIs), longer phases of arrhythmia were marked and excluded from the analyses (i.e., set to missing). In total, we removed per patient and daytime on average 0.3% (*min* = 0%, *max* = 7.2%) of the data points (for the amount of individual patients’ missing data, see Appendix A). Importantly, we paid special attention to adequate artefact correction because of the susceptibility of ApEn to noisy data, outliers, and missed heart beat detection [27].

Patients’ HRV data were analyzed in the frequency, time, and entropy domains. Due to obvious non-stationarities in the IBI signals (i.e., sudden shifts in the mean and/or standard deviation) we used complex demodulation (CDM) for quantifying the HRV frequency parameters. CDM produces equivalent results to fast Fourier transform but is less affected by non-stationarity (a prerequisite for accurate fast Fourier transform) [33,34].

We computed the mean IBI oscillation amplitude (in ms) for the very low (VLF; 0.003–0.04 Hz), low (LF; 0.04–0.15 Hz), and high (HF; 0.15–0.4 Hz) frequency bands for each minute. During the export of the mean IBI oscillation amplitude, ANSLAB automatically interpolates segments that were set to missing. Thus, we post hoc excluded all segments that were automatically interpolated using R version 3.6.1. [35]. Mean IBI (in ms; i.e., the time interval between two successive R-peaks), mean HR (equivalent to 60,000/IBI; in beats/min), and mean RMSSD (in ms) were also computed for each minute. For statistical analyses of HR/HRV parameters, all one-minute segments within each 6-h period (i.e., 8 a.m.–2 p.m., 2 p.m.–8 p.m., and 11 p.m.–5 a.m.) were averaged using R. Thus, we arrived at one value per patient and HR/HRV parameter for forenoon, afternoon, and night. For quantifying the complexity of the patients’ HR variations, ApEn, detrended fluctuation analysis scaling exponent (DfaAlpha), sample entropy (SampEn), and the Hurst exponent (Hurst) were computed for the whole 6-h segments (i.e., forenoon, afternoon, and night). ApEn describes the likelihood that patterns remain similar for subsequent comparisons, with higher ApEn values indicating higher irregularity and complexity in time-series data. DfaAlpha quantifies the presence or absence of fractal correlation properties in non-stationary time-series data, with higher DfaAlpha values suggesting fractal-like HR dynamics. SampEn describes the probability that two sequences of 1–5 consecutive data points (i.e., SampEn1-5) that are similar to each other will remain similar when one consecutive point is included. Higher SampEn indicates frequent incidence of dissimilarities in the time-series data. The Hurst exponent is a quantification of long-range dependence, with values of 0.5–1 indicating a long-term positive correlation, meaning that a high value in the series will probably be followed by another high value, values of 0–0.5 indicating time series with long-term negative correlation or switching between high and low values, and a value of 0.5 suggesting that the series are uncorrelated. For more details on non-linear HRV analyses, see Refs. [27,28].

#### 2.3.4. Respiration

Since DOC may be related to respiratory alterations [36] and as low respiration rates (i.e., <9 cycles per minute (cpm) ≤ 0.15 Hz) can affect the HF estimates of HRV via shifts in the respiratory sinus arrhythmia to the LF band [37], we analyzed the respiratory signal in a subsample of patients (*n* = 12) for whom respiration was simultaneously recorded via a respiration belt around the thorax. Mean respiratory rate for each minute was computed in ANSLAB and averaged for 5-min segments using R. We found that only one 5-min segment (i.e., 1.39% of 6-h recordings) of two patients (i.e., P17: forenoon; P26: night) was <9 cpm. Thus, we decided that there was no need to control the HF parameter for respiration in our patient sample. Although respiration data were only available from a subset of patients, it is unlikely that breathing patterns in the other patients were appreciably different.

#### 2.3.5. EEG Permutation Entropy

For analyzing heart–brain interaction, we correlated HRV entropy with EEG entropy in the Belgian patient sample (*n* = 14) where PSG was recorded. As the EEG data of the Belgian sample were already used in a previous study, where EEG permutation entropy (PE) had been computed, we used the EEG entropy values of the respective patients from that publication [29]. More specifically, PE of the entire EEG signal was computed for day (i.e., 8 a.m.–8 p.m.) and night (i.e., 11 p.m.–5 a.m.). PE quantifies the level of irregularity or unpredictability of an EEG signal. Higher PE values indicate more complex and/or random signals. For further information on the preprocessing of the EEG signal and the entropy analyses, see Ref. [29].

### 2.4. Statistical Analyses

Statistical analyses were done in R. In a first step, we evaluated whether the distribution of the data followed a normal distribution. As this was not the case for several variables (i.e., Shapiro–Wilk test for normality: *p* < 0.001; cf. Appendix A), we opted for advanced semi-parametrical and non-parametrical statistical tests. Specifically, for the analyses of differences in HR/HRV parameters (i.e., IBI, HR, RMSSD, VLF, LF, HF, ApEn, DfaAlpha, Hurst, and SampEn) and EEG entropy (i.e., PE) between different times of the day (i.e., within-subjects factor; forenoon, afternoon, night), diagnoses (i.e., between-subjects factor; (E)MCS vs. UWS), etiology (i.e., between-subjects factor; TBI vs. NTBI), or sex (i.e., between-subjects factor; female vs. male), we used advanced semi-parametric analyses for repeated measures designs as implemented in the ‘MANOVA.RM’ package available for R [38]. We report the resampled Wald-type statistic (WTS). As a resampling method, the function ‘perm’ was used, which randomly permutes all observations. The number of iterations used to calculate the resampled statistic was 10,000. To correct for multiple tests, *p*-values of post hoc comparisons were adjusted using the method of Benjamini and Hochberg (BH) [39] as implemented in the ‘p.adjust’ function in R. For correlation analyses of HR/HRV parameters, CRS-R sum score, and EEG entropy, we report Kendall’s Tau.

The significance level was set to α = 0.05 (two-sided) for all analyses. As suggested by Wasserstein et al. [40], we interpreted the overall pattern rather than focusing on individual *p*-values. Therefore, we also interpreted *p*-values 0.05 < *p* ≤ 0.10 if they were in line with the overall pattern of results and also ‘credible’ from a Bayesian point of view. More specifically, we additionally tested all trends of our post hoc comparisons and correlation analyses with Bayesian multilevel regression models via the Stan-based ‘brms’ package available for R [41,42]. We report regression coefficients and the 95% credible intervals (CIs; i.e., Bayesian confidence intervals). The CI describes the interval in which a parameter value falls with a 95% probability given the data observed, prior and model assumptions. Thus, an effect is considered to significantly differ from zero if zero is not included in the CI. We used weak- or non-informative default priors whose influence on results is negligible. For all computed regression models, no divergent transitions occurred, Rhat (i.e., potential scale reduction factor on split chains) was < 1.01, and effective sample size (ESS) was > 400.

## 3. Results

### 3.1. Interbeat Interval and Heart Rate

Analyses of the IBIs of 26 patients revealed a trend towards a main effect for *time* (*F_WTS_*(2) = 6.52, *p* = 0.068) and a significant effect for *diagnosis* (*F_WTS_*(1) = 5.8, *p* = 0.028) but no significant *time*
 × 
*diagnosis* interaction (*F_WTS_*(2) = 1.36, *p* = 0.523). Specifically, the patients showed larger IBIs during the night as compared to forenoon (*F_WTS_*(1) = 7.76, *p* = 0.027) and afternoon (*F_WTS_*(1) = 4.74, *p* = 0.059; *b* = −37.44, 95%CI = [−67.46, −7.42]). No differences could be observed in the patients’ IBIs between fore- and afternoon (*F_WTS_*(1) = 0.21, *p* = 0.648; cf. Figure 1a). Further, patients in UWS showed larger IBIs as compared to patients in (E)MCS (*F_WTS_*(1) = 5.8, *p* = 0.028; cf. Figure 1b).

This effect was supported by correlation analyses of IBIs and CRS-R sum scores of 25 patients showing that lower CRS-R sum scores were associated with larger IBIs during forenoon (*r*τ(23) = −0.34, *p* = 0.02; cf. Figure 2a) and afternoon (*r*τ(23) = −0.38, *p* = 0.009; cf. Figure 2b) and by trend during the night (*r*τ(23) = −0.27, *p* = 0.07; cf. Figure 2c). However, we refrain from interpreting the effect during the night as it does not appear robust from a Bayesian point of view (*b* = −12.34, 95%CI = [−25.79, 1.16]).

Analyses of HR yielded similar results (cf. Appendix A), which is expected as HR is inversely proportional to the IBI signal. In both IBI and HR, no differences for etiology (i.e., TBI vs. NTBI) or sex (i.e., male vs. female) were observed (cf. Appendix A).

### 3.2. HRV Time Domain

Analyses of the RMSSD of 26 patients did not reveal significant main effects for *time* (*F_WTS_*(2) = 0.4, *p* = 0.828), *diagnosis* (*F_WTS_*(1) = 0.37, *p* = 0.554), and *time*
 × 
*diagnosis* interaction (*F_WTS_*(2) = 3.91, *p* = 0.174; cf. Figure 3). No differences for etiology (i.e., TBI vs. NTBI) or sex (i.e., male vs. female) were observed (cf. Appendix A).

### 3.3. HRV Frequency Domain

Analyses of the VLF of 26 patients yielded a trend towards a main effect for *time* (*F_WTS_*(2) = 6.63, *p* = 0.064) but no significant effect for *diagnosis* (*F_WTS_*(1) = 1.74, *p* = 0.207) and the *time*
 × 
*diagnosis* interaction (*F_WTS_*(2) = 1.08, *p* = 0.603). Post hoc comparisons of VLF between times of day did not yield significance anymore after correcting for multiple comparisons (i.e., fore- vs. afternoon: *F_WTS_*(1) = 1.15, *p* = 0.295; forenoon vs. night: *F_WTS_*(1) = 1.26, *p* = 0.295; afternoon vs. night: *F_WTS_*(1) = 4.73, *p* = 0.111 (before BH-correction: *p* = 0.03); cf. Figure 4a).

Analyses of the LF of 26 patients revealed a trend towards a main effect for *time* (*F_WTS_*(2) = 5.69, *p* = 0.088) but no significant effect for *diagnosis* (*F_WTS_*(1) = 0.46, *p* = 0.51) and the *time*
 × 
*diagnosis* interaction (*F_WTS_*(2) = 0.36, *p* = 0.843). Post hoc comparisons of LF between times of day did not yield significance anymore after correcting for multiple comparisons (i.e., forenoon vs. afternoon: *F_WTS_*(1) = 1.65, *p* = 0.213; forenoon vs. night: *F_WTS_*(1) = 1.75, *p* = 0.213; afternoon vs. night: *F_WTS_*(1) = 4.86, *p* = 0.108 (before BH-correction: *p* = 0.028); cf. Figure 4b).

Analyses of the HF of 26 patients did not reveal significant main effects for *time* (*F_WTS_*(2) = 3.8, *p* = 0.191), *diagnosis* (*F_WTS_*(1) = 0.57, *p* = 0.460), and the *time*
 × 
*diagnosis* interaction (*F_WTS_*(2) = 0.04, *p* = 0.980; cf. Figure 4c).

No etiology or sex differences were observed for any frequency band (cf. Appendix A).

### 3.4. HRV Entropy Domain

Analyses of the ApEn of 26 patients revealed a significant main effect for *time* (*F_WTS_*(2) = 21.35, *p* = 0.001) but no significant effect for *diagnosis* (*F_WTS_*(1) = 1.79, *p* = 0.197) and the *time*
 × 
*diagnosis* interaction (*F_WTS_*(2) = 1.99, *p* = 0.395). Specifically, the patients showed a higher ApEn during forenoon as compared to afternoon (*F_WTS_*(1) = 15.79, *p* < 0.001). No differences could be observed in the patients’ ApEn during the night as compared to forenoon (*F_WTS_*(1) = 2.19, *p* = 0.154) and afternoon (*F_WTS_*(1) = 2.77, *p* = 0.154; cf. Figure 5). Analyses of other entropy parameters did not yield significant main effects for *time* and *diagnoses* and the *time*
 × 
*diagnosis* interaction (cf. Appendix A).

However, analyses of the DfaAlpha and SampEn1 yielded a significant main effect for etiology, with higher DfaAlpha (*F_WTS_*(1) = 4.59, *p* = 0.043; cf. Figure 6a) and SampEn1 (*F_WTS_*(1) = 4.36, *p* = 0.049; cf. Figure 6b) being observed in the patients with TBI as compared to patients with NTBI. No etiology and sex differences were observed for the other entropy parameters (cf. Appendix A).

### 3.5. Correlation of EEG and HRV Entropy

Correlation analyses in 14 patients showed that a higher ApEn was associated with a higher PE during the night (i.e., 11 p.m.–5 a.m.; *r*τ(12) = 0.47, *p* = 0.019; cf. Figure 7b). No significant correlation was observed during the day (i.e., 8 a.m.–8 p.m., *r*τ(12) = 0.3, *p* = 0.157, cf. Figure 7a).

## 4. Discussion

In patients with DOC, variations in cardiac activity show a diurnal pattern. Specifically, we find preserved diurnal variations in the length of the patients’ IBIs. The IBIs were larger during the night than during the day, indicating that, as in healthy individuals [22], the heart slows down during the night due to parasympathetic dominance reflecting relaxation and sleep. Further, the complexity of patients’ HRV signal varies across wakefulness, with the signal being more irregular and complex (i.e., higher ApEn) during forenoon as compared to afternoon. This has also been found in healthy individuals [43] and might be due to an increased cardiac sympathovagal response in the morning after awakening. More specifically, it has been shown in healthy individuals that sleep-to-wake transitions in the morning are associated with higher sympathetic activation compared to those occurring during the rest of the day [22]. Thus, although patients with DOC often fluctuate between sleep and wake-like phases—hence experiencing several ‘awakenings’ throughout the day—the results indicate that the awakening in the morning is probably the one associated with the most prominent change in arousal. Further, it might also be the case that patients show less frequent or temporally more regular (i.e., systematic) alterations of arousal and/or awareness in the afternoon, which may be associated with lower entropy values. This would go in line with our findings from a previous study, where the patients tended to exhibit higher CRS-R sum scores at a later daytime (i.e., afternoon) [8], which also requires more stable arousal and awareness levels. Another explanation for the entropy drop from fore- to afternoon might be the recovery from stressful events that possibly took place more frequently during forenoon (e.g., therapies, medical rounds, nursing activities). It has been shown in healthy controls that a stressful task leads to a significant reduction in entropy in the succeeding relaxation period [44]. Thus, the variation in cardiac activity over the day might be necessary for an optimal interaction and adaptation to changing demands in the environment. Interestingly, while we found that the patients in UWS generally had larger IBIs (i.e., lower HRs) than the patients in (E)MCS, lower IBIs were associated with higher behavioral reactivity (i.e., higher CRS-R sum scores) in the patients with DOC only during the day but not during the night. Looking at the data, one can see that this effect is mainly driven by the nocturnal slowing of the heart in the patients in an (E)MCS, underlining that cardiac activity shows a diurnal pattern, particularly when patients have (partially) regained consciousness. Thus, our findings complement earlier research suggesting that better circadian rhythm integrity is associated with higher consciousness levels [8,23,24]. More specifically, we found in earlier studies that variations in peripheral biosignals, such as skin temperature, melatonin(-sulfate), and wrist actimetry, are better aligned to a healthy 24-h rhythm (i.e., circadian rhythm) and more pronounced in patients with a higher behavioral repertoire [8,23,24].

Concerning central biosignals, such as brain activity derived from EEG recordings, earlier studies have shown that more severely affected patients with UWS do not only show a stronger general slowing of the EEG but also no clear diurnal pattern [29,45]. This is in line with the findings we present here (i.e., lower HR/larger IBIs in the patients with UWS) and possibly reflects the interaction between peripheral and central biosignals. Thus, we additionally investigated whether there is an association between the heart and central measures of brain activity, that is, EEG entropy (i.e., a measure describing the level of irregularity or unpredictability of the brain signal). We found that heart and brain activity are coupled during the night but not during the day. Specifically, while during the night a higher EEG entropy (i.e., PE) was associated with a higher ECG entropy (i.e., ApEn), no such association was evident during the day. It might be the case that, during the night (i.e., habitual sleep), and due to less disturbance by external cues from the environment and a stronger focus on internal processes, brain and body rhythms are better connected. Conversely, during the day, the brain focuses more on the processing of sensory signals in the environment, leading to a decrease in the synchrony between brain and body rhythms. Interestingly, the effect in our data seems to be mainly driven by EEG entropy. Specifically, the patients showed lower EEG entropy values during the night as compared to the day (see Appendix A: EEG Entropy), which might be an effect of the dominance of slow oscillations during the night (i.e., more synchronized brain activity, and thus less signal complexity) and probably indicates the existence of a ‘sleep-like state’ during the night. ECG entropy, however, does not differ between day and night. A reason for this could be that the patients spend most of the time in a lying position or a position where the upper body is raised in a 45° angle, which reduces events that usually influence HRV, such as changes in posture or physical movements. The brain signal, however, can still change independently—based on the patients’ clinical state. Specifically, while the patients in an (E)MCS showed higher EEG complexity during the day than during the night, no diurnal variation was evident in the patients with UWS (see Appendix A: EEG Entropy).

Additionally, cardiac activity does not only inform regarding the integrity of diurnal variations and the degree of behavioral reactivity in patients with DOC but also differentiates between patients’ etiologies. Specifically, the patients with TBI showed a higher DfaAlpha and SampEn1 than the patients with NTBI, suggesting fractal-like (i.e., aperiodic) HR dynamics and higher dissimilarities in the HRV signal of the patients with TBI. In other words, cardiac activity is less complex and variable in patients with NTBI. Reduced HRV complexity has been shown to be a predictor for mortality [46,47]. This is in line with previous findings showing that patients with NTBI often have a less favorable prognosis than patients with TBI [48,49,50]. Importantly, while this study serves as a proof of principle study, additional clinical and multicenter studies are needed that investigate the extent of diagnostic quality (i.e., in terms of sensitivity and specificity) of the different HRV parameters.

When looking at the time (i.e., RMSSD) and frequency (i.e., VLF, LF, HF) domains of HRV, no differences between time of day, diagnoses, and etiologies were evident. One reason might be of a methodological nature. More specifically, the activity of the heart is not regular/periodic but rather fluctuates in complex/aperiodic patterns. Thus, it has frequently been argued that non-linear measures (i.e., measures of mathematical chaos/entropy) might be more appropriate for the analysis of HRV data [51,52]. This is in line with the findings from a study that showed a difference in HRV entropy between patients with UWS and healthy individuals (i.e., lower ApEn in patients with UWS) but no such differences in any of the linear parameters (i.e., IBI, SDRR, RMSSD, LF/HF ratio) [18].

Our results show that ApEn is of diagnostic relevance in our study sample. However, due to its special requirements and limitations, it has been discussed whether it is an ideal measure for quantifying entropy [27,53]. In our study, the often instead used ‘SampEn’ measure does not reveal the same results as ApEn (cf. Appendix A). A possible reason might be that ApEn correlates with RMSSD and HF power (see Table 1 in Ref. [27]), while SampEn correlates, rather, with LF power. ApEn thus combines aspects of HRV time and frequency domain measures with the nonlinear measure ‘entropy’ in a way that seems to be of particular diagnostic relevance for our research questions in severely brain-injured patients.

## 5. Conclusions

To summarize, patients with severe brain injuries—particularly those who (partially) regained consciousness—still had preserved diurnal variations, as characterized by a heart rate slowing during the night. This suggests preserved integrity of circadian rhythms in the autonomic nervous system activity. Further, cardiac activity differentiated between patients’ etiologies and diagnoses. The patients with UWS had larger IBIs (i.e., lower heart rate) than the patients in an (E)MCS, and the patients with NTBI had a less complex HRV signal than the patients with TBI. Thus, cardiac activity and its variations might represent a peripheral window to central (brain) functioning. Indeed, we found an interaction of heart and brain signal complexity, which also followed a diurnal pattern. Specifically, while a more complex brain signal was associated with a more complex heart signal during the night, no such association was found during the day. In conclusion, HR and HRV seem to mirror the integrity of brain functioning and, consequently, might serve as supplementary measures that aid the differentiation between clinical states. Ultimately, this has the potential to improve the validity of assessments in patients with DOC.

## Figures and Tables

**Figure 1 brainsci-12-00375-f001:**
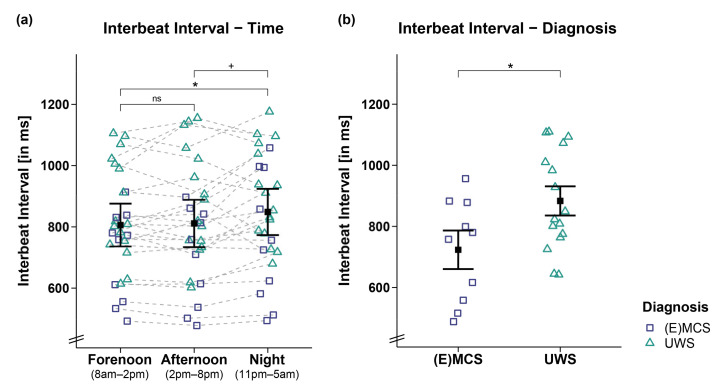
Interbeat interval (IBI) separately for time and diagnosis contrasts. (**a**) While patients’ IBIs were larger during the night as compared to the day (i.e., forenoon, afternoon), they did not differ between fore- and afternoon. (**b**) Patients in UWS showed larger IBIs than patients in (E)MCS. Error bars represent the mean and 95% confidence interval. * *p* < 0.05; ^+^
*p* ≤ 0.1; ns = not significant. Abbreviations: (E)MCS = (emergence from) minimally conscious state; UWS = unresponsive wakefulness syndrome; ms = milliseconds.

**Figure 2 brainsci-12-00375-f002:**
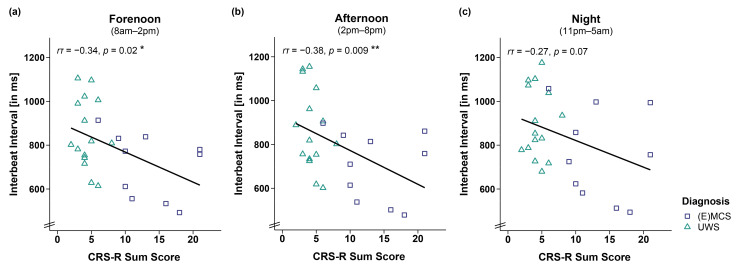
Correlation between interbeat interval (IBI) and CRS-R sum score separately for time. Larger IBIs were associated with lower CRS-R sum scores throughout the (**a**,**b**) day (i.e., forenoon, afternoon) and (**c**) night. Please note that the effect during the night is no longer ‘credible’ from a Bayesian point of view, and will not be interpreted. ** *p* < 0.01; * *p* < 0.05. Abbreviations: (E)MCS = (emergence from) minimally conscious state; UWS = unresponsive wakefulness syndrome; ms = milliseconds.

**Figure 3 brainsci-12-00375-f003:**
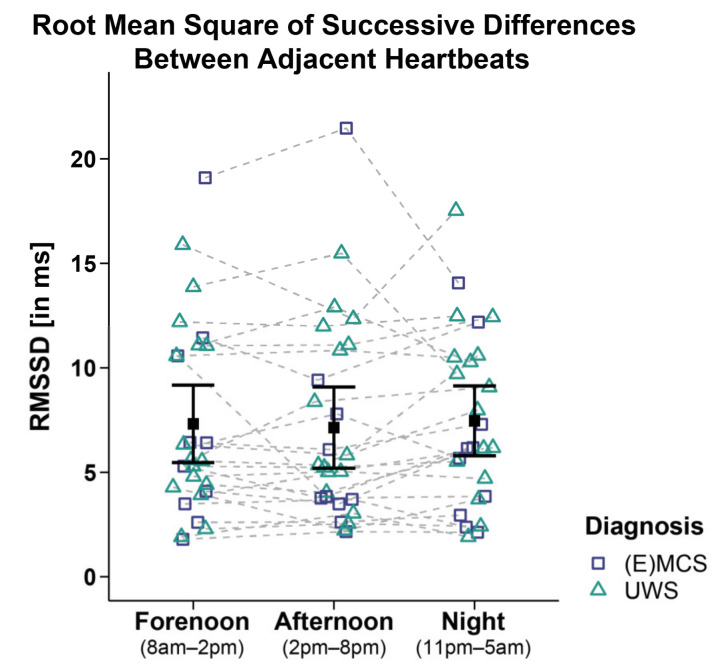
Root mean square of successive differences between adjacent heartbeats (RMSSD). Patients’ RMSSD did not differ between time ranges and diagnoses. Error bars represent the mean and 95% confidence interval. Abbreviations: (E)MCS = (emergence from) minimally conscious state; UWS = unresponsive wakefulness syndrome; ms = milliseconds.

**Figure 4 brainsci-12-00375-f004:**
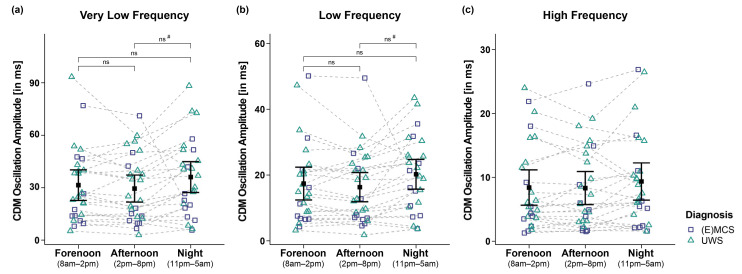
Very low (VLF), low (LF), and high frequency (HF) oscillation amplitude of the IBI signal computed by complex demodulation (CDM). Analyses of the (**a**) VLF and (**b**) LF band revealed significant main effects for time (i.e., forenoon, afternoon, night). However, post hoc comparisons between times of day did not yield significance anymore after correction for multiple comparisons. (**c**) Patients’ HF also did not differ between times and diagnoses. Error bars represent the mean and 95% confidence interval. # = significant before correction for multiple comparisons; ns = not significant. Abbreviations: (E)MCS = (emergence from) minimally conscious state; UWS = unresponsive wakefulness syndrome; ms = milliseconds.

**Figure 5 brainsci-12-00375-f005:**
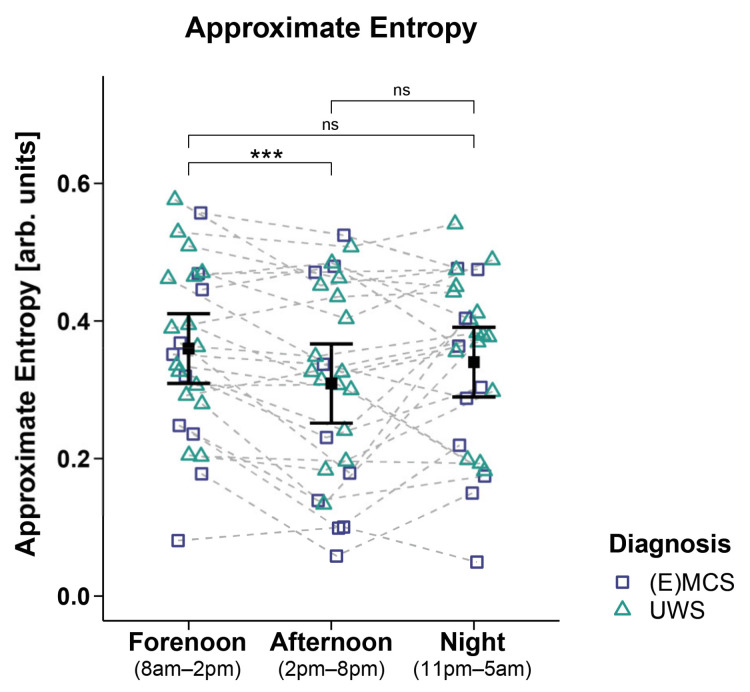
Approximate entropy (ApEn). Patients showed a higher ApEn during forenoon as compared to afternoon. No significant differences were evident between night and day (i.e., night vs. forenoon or afternoon). Error bars represent the mean and 95% confidence interval. *** *p* < 0.001; ns = not significant. Abbreviations: (E)MCS = (emergence from) minimally conscious state; UWS = unresponsive wakefulness syndrome; arb. units = arbitrary units.

**Figure 6 brainsci-12-00375-f006:**
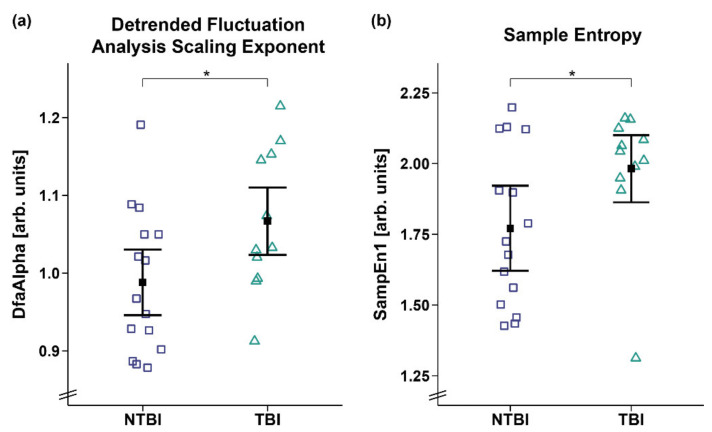
Detrended fluctuation analysis scaling exponent (DfaAlpha) and sample entropy 1 (SampEn1). Patients with TBI showed a higher (**a**) DfaAlpha and (**b**) SampEn1 as compared to patients with NTBI. Error bars represent the mean and 95% confidence interval. * *p* < 0.05. Abbreviations: NTBI = non-traumatic brain injury; TBI = traumatic brain injury; arb. units = arbitrary units.

**Figure 7 brainsci-12-00375-f007:**
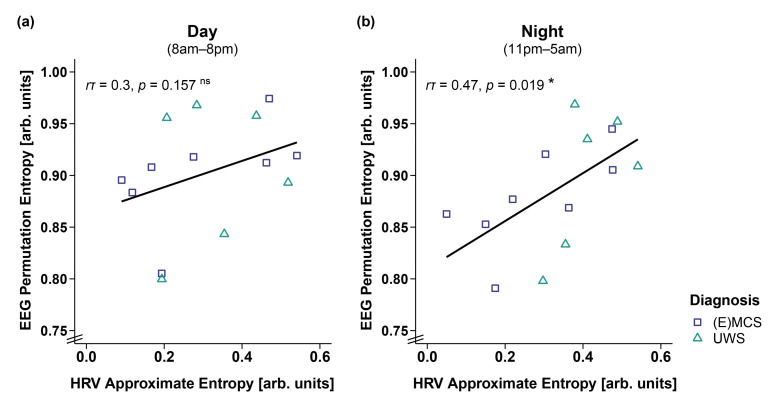
Correlation between EEG permutation entropy (PE) and HRV approximate entropy (ApEn) separately for day and night. (**a**) While EEG PE and HRV ApEn did not correlate during the day (i.e., 8 a.m.–8 p.m.), (**b**) a higher EEG PE was associated with a higher HRV ApEn during the night (i.e., 11 p.m.–5 a.m.). * *p* < 0.05; ns = not significant. Abbreviations: (E)MCS = (emergence from) minimally conscious state; UWS = unresponsive wakefulness syndrome; arb. units = arbitrary units.

**Table 1 brainsci-12-00375-t001:** Demographic information and highest CRS-R sum score/diagnosis.

Patient ID	Age	Sex	Etiology	Time Since Injury (Months)	Diagnosis	CRS-R Sum Score
P1	64	M	NTBI	8.8	UWS	3
P2	77	M	NTBI	15.7	UWS	3
P3	35	M	NTBI	14.6	UWS	4
P4	71	M	NTBI	20.7	EMCS	21
P5	55	F	NTBI	23.8	UWS	4
P6	59	M	NTBI	25.8	UWS	4
P7	80	M	TBI	21.9	MCS	9
P8	22	F	NTBI	48.3	UWS	-
P9	48	M	TBI	7.7	UWS	5
P10	76	M	TBI	3.5	UWS	4
P11	70	M	NTBI	3.4	UWS	2
P12	71	F	TBI	4.7	UWS	4
P13	16	F	TBI	0.8	MCS	16
P14	21	M	TBI	7.0	UWS	6
P15	48	M	NTBI	1.4	MCS	18
P16	66	M	NTBI	3.2	MCS	10
P17	61	M	NTBI	2.0	MCS	10
P18	36	M	TBI	6.0	MCS	6
P19	74	F	TBI	0.5	UWS	3
P20	31	F	NTBI	1.4	MCS	11
P21	43	F	TBI	6.0	MCS	21
P22	61	F	NTBI	0.9	UWS	6
P23	37	M	NTBI	9.4	UWS	5
P24	34	M	NTBI	240.0	UWS	8
P25	32	M	TBI	6.1	UWS	5
P26	20	M	TBI	36.0	MCS	13

M = male; F = female; NTBI = non-traumatic brain injury; TBI = traumatic brain injury; UWS = unresponsive wakefulness syndrome; MCS = minimally conscious state; EMCS = emergence from MCS; CRS-R = Coma Recovery Scale-Revised (cf. Behavioral Assessment and Data Analysis). Please note that we could not obtain valid CRS-R assessments in one patient (P8) because not all subscales could be evaluated (i.e., due to eyes being closed and it being impossible to induce eye-opening even when physically stimulating the patient). In this case, we used the diagnosis that was obtained during a CRS-R assessment ten days earlier. While the diagnosis usually stays quite stable over time, CRS-R scores can slightly change. Thus, we did not use the CRS-R sum score of this earlier assessment for analyses.

## Data Availability

The data that support the findings of this study are available from the corresponding authors upon reasonable request.

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
