# Peer review of "Does the Heart Fall Asleep?—Diurnal Variations in Heart Rate Variability in Patients with Disorders of Consciousness"

_brainsci, 2022, doi:10.3390/brainsci12030375_

Round 1

Reviewer 1 Report

The paper reports on hrv differences between TBI and NTBI patients. It is found that TBI patients are more liable to have severe complications on recovery. This is an interesting paper, however, I have the following problems with the it:

  • It would seem that the Authors use statistical methods without looking at their data. e.g. fig.2 and fig.7 are severely scattered and so running a linear regression does not make sense. In other figures, one can see a large overlap of the data in the different categories. Thus comparing averages and percentiles does not make sense - they come out statistically significant but only in the sense of sensitivity. And what about the specificity of the results? This should also be calculated and discussed. The overlap visible in the figures means that the method proposed by the Authors will most probably mean a very low specificity.
  • Finally, the Authors seem to rely on ApEn which has been shown to be wrong by Richmann and Moormann [17]. They introduced SampEN to be used instead of ApEn and they published the reasons in ref. [17] of the paper.
  • details:
  • fig.3 caption: 'between times' => between time ranges

Reviewer 2 Report

The authors explored the difference in heart rate and heart rate variability between patients in Unresponsive Wakefulness Syndrome (UWS) and patients in/emerging from Minimally Conscious State (MCS/eMCS). They also check for any correspondence between the autonomic measures and brain activity in a subsample of patients. Overall, results show the preservation of the diurnal variation of heart signal along with an association between brain and heart signals during the night. Furthermore, the interbeat interval discriminates between UWS and eMCS/MCS, while the HRV signal better discriminates between TBI and non-TBI.  

I found the introduction clear and well-written. I only have minor concerns referring to this section:

  • Page 2, line 53: I suggest to add “at least” in the bracket before minimally: “Thus, while patients with UWS are assumed to be unconscious, patients in MCS and EMCS are assumed to be (at least minimally) conscious”.
  • Page 2, line 57: I suggest to add a reference after the following sentence: “Unfortunately, behavioral assessments involve the risk of underestimating the level of consciousness”.
  • Page 2, line 63: I suggest the following minor change to the sentence “Hence, distinguishing between UWS and (E)MCS continues to be a challenge in clinical practice, and the rate of misdiagnoses is high (i.e., ~40%) [8] when comparing the medical consensus to the results obtained through clinical scales”

For the materials and methods section, I have some minor and major comments:

  • Page 3, line 115: please, report the mean and standard deviation of the age of patients instead of the median value.
  • Page 3, line 119: the authors stated “MCS and EMCS patients were combined for statistical analyses” however, I do not agree with this, unless the authors prove that no statistical difference exists between the EMCS patient profile and the rest of the group (MCS patients). Alternatively, the authors should provide a theoretical reason behind this choice. Indeed, patients in MCS and those who are emerging from MCS represent two different clinical categories both from a behavioural and functional point of view. For these reasons, I do not see any theoretical reason behind the authors’ choice to consider together these clinical categories.
  • Page 3, lines 120-122: If I understood correctly, this is a retrospective study with data collected for different purposes. I suggest to clearly state that it is a retrospective study.
  • Page 3, table 1: Here, or in the supplementary materials, I suggest to give to the readers details about the sub-scores obtained by each patient to the CRS-r subscales.
  • Page 4, line 142: Please, specify what “biodynamic” means.
  • Page 5, lines 155-158: In Austria, the study protocol comprises a continuous registration along the week. In Belgium, instead, there was a 24-hours registration. It is not clear to me which data have been considered from the Austrian protocol (was it the first 24-hours registration during the HL week for each patient?). To be honest, further on (page 5, lines 186-187), the authors stated “we used continuous 24-h ECG data from each patient”. I suggest to better specify if they selected the first 24-h registration from the Austrian dataset.
  • Overall, I am concerned about considering data registered with two different instruments. Can the authors provide evidence of the absence of difference depending on which instrument has been used to register patients’ data?
  • Please, report the extended form of the “IBI” acronym at its first appearance in the main text.

I further suggest to cite these works for discussion purposes:

Leo A, Naro A, Cannavò A, Pisani LR, Bruno R, Salviera C, Bramanti P, Calabrò RS. Could autonomic system assessment be helpful in disorders of consciousness diagnosis? A neurophysiological study. Exp Brain Res. 2016 Aug;234(8):2189-99. doi: 10.1007/s00221-016-4622-8. Epub 2016 Mar 25. PMID: 27016088.

Riganello F, Napoletano G, Cortese MD, Arcuri F, Solano A, Lucca LF, Tonin P, Soddu A. What impact can hospitalization environment produce on the ANS functioning in patients with Unresponsive Wakefulness Syndrome? - 24-hour monitoring. Brain Inj. 2019;33(10):1347-1353. doi: 10.1080/02699052.2019.1641841. Epub 2019 Jul 23. PMID: 31335209.

Round 2

Reviewer 1 Report

Due to the large overlap of the results, the 'basic research' answer does not sound real.  Moreover, due to this overlap it would really be good to see what are the specificity etc. 

Also, ApEn (although still used by many authors) has been shown to be bad. Because of this SampEn should be used instead. The fact that ApEn gives good results is no excuse.

I have not changed my view that the (extended) statistical methods used by the Authors are not correct. The comparison between the groups is based on a comparison of averages in the two groups. That is not enough.

I move to reject this paper.

Reviewer 2 Report

I do not have further comments. I am satisfied with the authors' responses to my questions.
